# Geometry-aware Two-scale PIFu Representation for Human Reconstruction

**Zheng Dong**[1]  **Ke Xu**[2]  **Ziheng Duan**[1]  **Hujun Bao**[1]  **Weiwei Xu**[*1]  **Rynson W.H. Lau**[2]

[1]State Key Lab of CAD&CG, Zhejiang University    [2] City University of Hong Kong

## Abstract

Although PIFu-based 3D human reconstruction methods are popular, the quality of recovered details is still unsatisfactory. In a sparse (*e.g.*, 3 RGBD sensors) capture setting, the depth noise is typically amplified in the PIFu representation, resulting in flat facial surfaces and geometry-fallible bodies. In this paper, we propose a novel geometry-aware two-scale PIFu for 3D human reconstruction from sparse, noisy inputs. Our key idea is to exploit the complementary properties of depth denoising and 3D reconstruction, for learning a two-scale PIFu representation to reconstruct high-frequency facial details and consistent bodies separately. To this end, we first formulate depth denoising and 3D reconstruction as a multi-task learning problem. The depth denoising process enriches the local geometry information of the reconstruction features, while the reconstruction process enhances depth denoising with global topology information. We then propose to learn the two-scale PIFu representation using two MLPs based on the denoised depth and geometry-aware features. Extensive experiments demonstrate the effectiveness of our approach in reconstructing facial details and bodies of different poses and its superiority over state-of-the-art methods.

## 1   Introduction

Three-dimensional human reconstruction, which aims to obtain a dense surface geometry from single-view or multi-view human images, is a fundamental topic in computer vision and computer graphics. While reconstructing high-fidelity 3D human models is possible using commercial multi-view/stereo software under the customized studio setting [14, 58, 81, 29, 85], it is highly desirable to lift the studio setting constraint, which may be inaccessible to most users. Low-cost RGBD sensors have recently become popular in 3D human reconstruction, and tracking-based methods are developed to fuse the depth data from RGBD sensors for reconstruction [66]. During fusion, the estimation of non-rigid human body deformation is essential to improve the reconstruction quality [65, 101]. However, it is technically challenging to ensure the stability of the depth fusion algorithm, due to occlusions and severe noise in the depth data.

Recently, learning-based 3D human reconstruction methods have significantly simplified the capture setting. The parametric human model [60, 69, 67] is introduced to reduce the modeling difficulties from complex poses. After training on images-to-model pairs, methods [46, 105] may even reconstruct 3D human shapes from single images. However, these methods typically only obtain minimally clothed human bodies. As detail requirements increase, the focus of learning-based human reconstruction methods has been shifted to the implicit representation, *e.g.*, pixel-aligned implicit function (PIFu) [73, 74, 40]. PIFu-based methods [73] can reconstruct human bodies with different types of details (*e.g.*, hair and clothing) without utilizing predefined templates. However, they often produce topology errors in the reconstructed human models, especially in the regions

---

[*]Corresponding author.

36th Conference on Neural Information Processing Systems (NeurIPS 2022).

that are invisible or where the input depth is highly noisy (*e.g.*, hairs). While PIFuHD [74] partially alleviates this problem by synthesizing the body's normal maps at both front and back sides, the details reconstructed from the synthesized normals may not be consistent with the target. Hence, some methods [40, 100] resort to the multi-view feature fusion scheme to reduce these topology errors.

Although implicit methods develop fast, we observe that the quality of their recovered details is still unsatisfactory under the sparse capture settings (*e.g.*, 3 RGBD sensors). Due to the less overlapping between sparse views, the input depth noise is typically amplified, increasing the difficulty of performing stable reconstruction. In addition, due to the ineffective fusion of RGB and depth features, these methods may not easily reconstruct high-frequency details. As a result, we often observe flat or incorrect facial surfaces, body geometries with topology errors, as shown in Fig. 1(b,c,d).

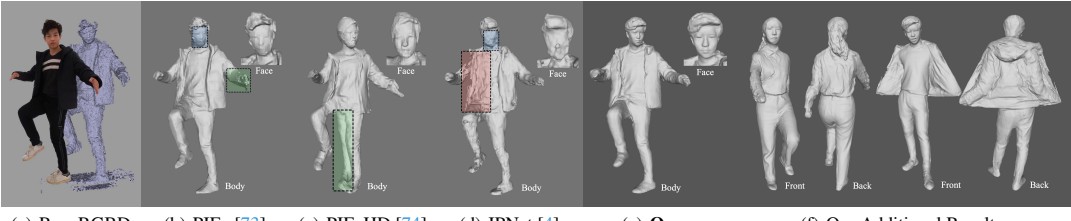

| (a) Raw RGBD | (b) PIFu [73] | (c) PIFuHD [74] | (d) IPNet [4] | (e) **Ours** | (f) Our Additional Results |

Figure 1: Given sparse and noisy inputs, with one of the three RGBD views shown in (a), existing methods such as Multi-view RGBD-PIFu [73] (b), PIFuHD [74] (c) and IPNet [4] (d) tend to produce over-smoothed facial details (b,d) or topology errors (c,d), due to the amplified noise in sparse views. Our method learns the geometry-aware two-scale PIFu representation, which can produce vivid facial/hair details and accurate bodies under different poses (e,f).

In this paper, we propose a geometry-aware two-scale PIFu representation for reconstructing digital humans from sparse, noisy inputs. Our method is based on two observations. First, depth denoising and 3D reconstruction are complementary to each other. The former task preserves local geometric fidelity, while the latter task provides global topology guidance. Second, while a function of high complexity may not be easy to express, it is much easier to approximate it piecewise (*e.g.*, in two parts). Inspired by these observations, we first formulate depth denoising and 3D reconstruction as a multi-task learning problem. The two tasks can work together to further improve the reconstruction quality. We encode RGB and depth separately, and use fused features to perform depth denoising, enriching deep features with local geometric information. Based on the denoised depth and geometry-aware features, we propose using two MLPs to represent the PIFus for the face (particular region) and the body, respectively. This separate modeling increases the network capacity for handling details at different scales. As shown in Fig. 1(e, f), our method can produce results with high-fidelity body and facial details for different actions. Our main contributions are:

**1)** We propose a novel geometry-aware PIFu method for digital human reconstruction from sparse, noisy RGBD images. Our approach exploits the complementary properties of depth denoising and 3D reconstruction to learn robust geometry information while suppressing the input noise.

**2)** We propose to learn a two-scale PIFu representation based on geometry-aware features, by using two individual MLPs for the face and the body separately. The two-scale formulation enhances the network capacity in producing high-frequency facial details and smooth body surfaces.

**3)** Extensive experiments demonstrate that the proposed method can produce high-quality digital human reconstruction results, based on noisy depth maps taken by three Azure Kinect sensors.

## 2 Related Works

**Tracking-based Human Reconstruction** methods [50, 66, 109, 65, 30, 19, 49, 18, 101, 33, 79, 100] track human motions and infer the non-rigid deformations of the references to reconstruct 3d human mesh in a temporal fusion manner. The references are typically parameterized as the 3D poses/skeletons [24, 98, 99, 92, 101, 33, 34, 55] and/or parametric body models [7, 102, 101, 45] (*e.g.*,SMPL [60, 69, 45]). Some methods combine tracking with segmentation [59] or optical flow estimation [63] to help compute the references for reconstruction. To tackle the occlusion and large motion problems, some methods develop high-end systems for the dense capture of human

performance, consisting of a large number (up to 100) of RGBD sensors [14, 58, 81] or custom color lights [29, 85] (*e.g.*, 1,200 individually controllable light sources in the acquisition setup in [85]).

**Learning-based Human Reconstruction** methods [108, 77, 2, 23, 88, 64, 106, 83, 93, 100] leverage the neural 3D representation for reconstructing the geometry and/or texture details. Some methods obtain 3D human meshes from a single RGB [77] or RGBD [88] image by incorporating the SMPL template [80], which are ineffective for deformed human bodies. Some approaches reconstruct continuous results from RGB [91] or RGBD [107] videos, skeletal motions [32], but are limited by garments types and cannot obtain facial expression details. Other methods adopt the image-to-image translation pipeline to regress the 3D mesh via 2D estimations of intermediate textures [2], silhouettes [64], and depths [23], while some approaches jointly exploit 2D and 3D information, *e.g.*, body joints and per-pixel shading information [108], and 2D/3D poses and segmentation map [83]. These methods typically suffer from over-smoothed surfaces in occluded regions.

Recently, the pixel-aligned implicit function (PIFu) [73] has attracted much attention for 3D human reconstruction due to its effective implicit representation, and PIFuHD [74] estimates normal maps to reduce the geometry errors in the occluded regions based on PIFu. Due to its success, many methods promote PIFu with voxel-alignment [104, 40, 37], deformation field [38, 41], real-time approach [51], illumination [3], monocular fusion [54], sparse-view temporal fusion [100], or apply it for point clouds based human reconstruction [13, 4, 62]. Typically, PIFu-based methods lack fine facial details in the sparse capture setting due to its all-in-one implicit 3D representation learning. To tackle this problem, JIFF [9] employs the facial 3DMM [5] model as a shape prior, enhancing pixel-aligned features for detailed geometry information. However, their face reconstruction capacity is constrained by the 3DMM parameters, so that JIFF may not handle dense facial geometry well. In this paper, we propose the geometry-aware two-scale PIFu representation, which utilizes high-resolution RGB images to obtain the face geometric details and explicitly fuses the face and body occupancy fields for high-quality full-body reconstructed results.

**Depth Image Denoising** is essential for depth images captured from consumer-level depth sensors (*e.g.*, Azure Kinect). Traditional filtering-based methods [47, 10, 17, 71, 36, 76, 35] enhance depth data from various sensors. Color or infrared images are used to help depth image denoising [16, 96, 56, 68, 57, 22, 95, 61, 35]. These methods typically assume the intensity image to be edge-aligned with the depth image, and they tend to produce artifacts when this assumption is violated. Some methods [28, 75, 66, 15, 39, 11, 65, 31] fuse multiple frames to refine the depth images, but they tend to produce over-smoothed depth images.

Learning-based depth image denoising methods are proposed based on dictionary-learning [48] and deep-learning [27, 53, 52, 44, 93, 78]. Recently, Yan *et al.* [93] propose the DDRNet for deep denoising using fused geometry and color images. Their method tends to produce inconsistent results. Sterzentsenko *et al.* [78] propose a self-supervised method that leverages photometric supervision from a differentiable rendering model to smooth the depth noise. However, the photometric supervision lacks 3D geometry information and their approach tends to produce homogeneous results. In this paper, we propose to formulate depth denoising and 3D reconstruction as a multi-task learning problem. The global topology information facilitates the depth denoising significantly.

**Monocular Face Reconstruction** methods aim to reconstruct personalized faces from monocular data. The parametric face model, *e.g.*, 3D morphable model (3DMM) [6, 5, 72] or multi-linear blend shapes [8, 84, 25], are typically used in conventional methods. However, these methods may not reconstruct accurate or dense facial geometry. Deep-learning-based methods [42, 97, 21, 26] that incorporate face landmarks or the 3DMM model for face reconstruction also tend to produce coarse reconstructions results. Other methods [82, 43, 21, 70, 103] estimate the facial dense shape instead of the low-dimensional template parameters. All these approaches only reconstruct faces. In this paper, we apply the PIFu representation and extend it to two scales for modeling human bodies and faces. Our formulation enables producing vivid facial expressions with accurate body reconstructions.

## 3 Proposed Method

We propose the geometry-aware two-scale PIFu method, to address the problem of PIFu-based approaches that reconstruct flat facial and geometry-fallible body surfaces in the sparse capture setting. First, to handle the noise issue of the sparse capture setting, we propose to formulate the depth denoising and the body reconstruction processes in the multi-task learning manner to exploit their complementary properties. Second, although the face only occupies a small proportion of the whole

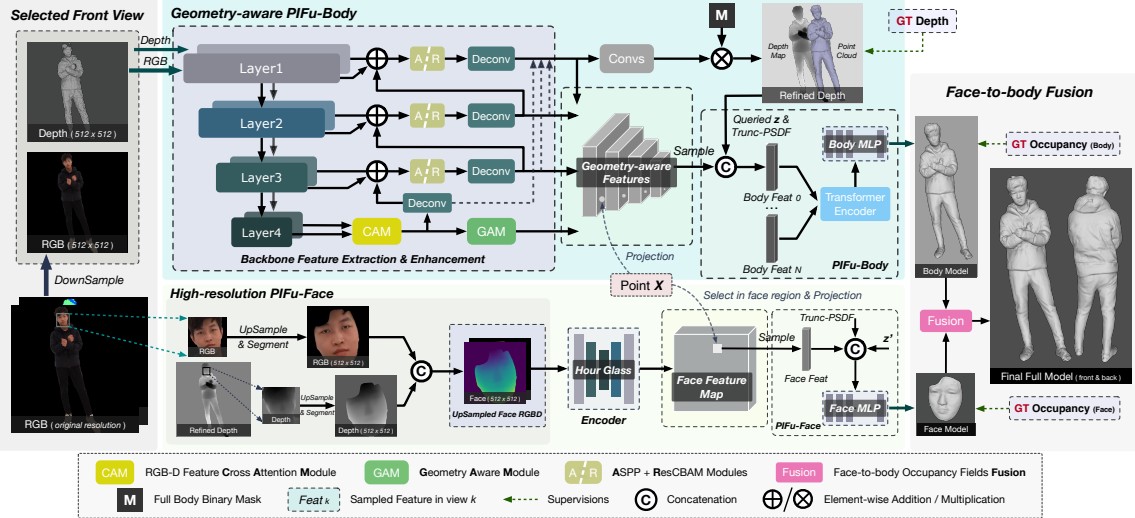

Figure 2: Proposed method overview. Given sparse and noisy RGBDs as inputs, *Geometry-aware PIFu-Body* performs depth denoising and predicts the body occupancy field. *High-resolution PIFu-Face* predicts the face occupancy field with fine-grained details. The body and face occupancy fields are fused to produce final results via the *Face-to-body Fusion* scheme.

human model, it typically contains more high-frequency (*e.g.*, vivid expressions) than other parts and plays a vital role in assessing the reconstruction fidelity. To this end, we propose the two-scale PIFu representation to allocate more network capacity for face reconstruction.

As illustrated in Fig. 2, our network contains three parts: (1) the *Geometry-aware PIFu-Body*, $\mathcal{F}_b$ that predicts the body occupancy field $O_b$ and the refined depth maps from the noisy RGBD images of $\mathcal{N}$ (*i.e.*, three) perspective views; (2) the *High-resolution PIFu-Face*, $\mathcal{F}_f$ which obtains the fine-grained face occupancy field $O_f$, using refined depth map from $\mathcal{F}_b$ and the high-resolution RGB image of the front view as inputs; (3) the light-weight *Face-to-body Fusion*, $\mathcal{W}$ that reconstructs the full human model by fusing the face and body occupancy fields (*i.e.*, $O_b$ and $O_f$).

## 3.1 Geometry-aware PIFu-Body: $\mathcal{F}_b$

PIFu-based methods predict 3D occupancy fields in an implicit manner, enabling image-aligned features to be aware of global topological information, which helps suppress depth noise. For example, in Fig. 3(e), the holes can be filled using only geometric supervision. However, this depth map tends to be over-smoothed, and the reconstructed surface also lacks details (Fig. 3(f)). On the other hand, with only depth supervision, image-to-image depth denoising can fill holes and add details. However, the obtained depth contains incorrect details (*e.g.*, face, hands, and clothing regions shown in Fig. 3(g) and Fig. 9(b)), when lacking of 3D geometric guidance. Based on these observations, we formulate depth denoising and PIFu-based occupancy estimation in a multi-task learning manner. It exploits the global topological information of the 3D occupancy field to guide the denoising process, and the local high-frequency details of refined depth to improve occupancy estimation (Fig. 3(h,i,j) and Fig. 9(c,h)).

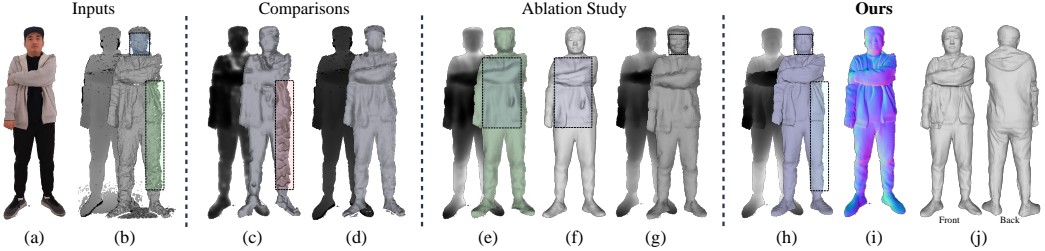

Figure 3: Visualization of depth denoising and the subsequent reconstruction results. Raw RGB and Depth images (a,b). Results of two depth denoising methods, [93] and [78] (c,d). Our results of only using occupancy supervision (e,f). Our refined depth under depth supervision only (g). Our refined depth, its normal map, and full-body mesh (h,i,j).

**Formulation of** $\mathcal{F}_b$. Given the triplet ($\mathbf{T}$) of RGB ($\mathbf{I}$), depth ($\mathbf{D}$) images, body binary masks ($\mathbf{M}$) from $\mathcal{N}$ perspective views, and the query point $\mathbf{X} \in \mathbb{R}^3$ as inputs, we formulate the $\mathcal{F}_b$ to predict both the body occupancy value $\sigma_b \in [0,1]$ and the refined depth maps $\mathbf{D}_{rf}$, as:

$$\mathcal{F}_b(\mathbf{X}, \mathbf{T}) = \{\mathcal{M}_b(\mathcal{A}(\{\mathcal{B}(\psi_g(\mathbf{T}_i), \mathbf{x}_i), \mathcal{C}_i(\mathbf{X})\}_{i=1,\dots,\mathcal{N}})), \mathcal{D}(\psi_g(\mathbf{T}_i))\} := \{\sigma_b, \mathbf{D}_{rf}^i\}, \quad (1)$$

where $\psi_g(\cdot)$ is a mapping function that encodes $\mathbf{T}$ into multi-scale Geometry-aware features, $\mathbf{x}_i = \pi_i(\mathbf{X}) \in \mathbb{R}^2$, is the 2D projection of point $\mathbf{X}$ at $i$-th view, and $\mathbf{z}_i \in \mathbb{R}$ is the depth of $\mathbf{X}$ in the local coordinate system of the $i$-th view. $\mathcal{C}_i(\mathbf{X}) = [\mathbf{z}_i, p_i(\mathbf{X})]$ where $p_i(\mathbf{X}) = \mathcal{T}(\mathbf{z}_i - \mathcal{B}(\mathbf{D}_{rf}^i, \mathbf{x}_i)) \in [-\delta_p, \delta_p]$, is the truncated PSDF value as in [100]. $\mathcal{B}(\cdot, \mathbf{x}_i)$ is the sampling function to obtain pixel-aligned 2D features $\mathcal{B}(\psi_g(\mathbf{T}_i), \mathbf{x}_i)$ and depth information $\mathcal{C}_i(\mathbf{X})$, which are processed by the multi-view feature aggregation module $\mathcal{A}$ and further fed into the implicit function $\mathcal{M}_b$ for occupancy querying. Meanwhile, the decoder $\mathcal{D}(\cdot)$ processes features $\psi_g(\mathbf{T}_i)$ for depth denoising.

**Geometry-aware Features:** $\psi_g(\mathbf{T}_i)$. The geometry-aware mapping $\psi_g(\cdot)$ aims to exploit the complementary properties of depth denoising and occupancy field estimation. Hence, it is expected to fuse the multi-modal RGB-D inputs effectively. To handle their modal discrepancy, we use two independent *HRNets* [87] (Fig. 2) to process the RGB ($\mathbf{I}$) and depth ($\mathbf{D}$) inputs respectively, where $\{\mathbf{I}, \mathbf{D}\}$ will be pre-processed to filter out the background via the masks, *i.e.*, $\mathbf{M}(\mathbf{I}, \mathbf{D})$. To fuse the RGB and depth backbone features, we propose a novel Cross Attention Module (CAM) on the highest level of backbone features, and use the CAM output feature to guide the fusion of lower levels in a level-wise manner. We also propose a novel Geometry Aware Module (GAM) to enrich the CAM output features with high-frequency information. The enhanced features form the features $\psi_g(\mathbf{T}_i)$.

**Cross Attention Module (CAM).** The CAM aims to fuse the RGB and depth features by computing their non-local correlations. Since depth is typically noisier than RGB information, we first compute the self-attention map based on the RGB feature and then use it to reweight both RGB and depth features before they are fused. The architecture of CAM is shown in Fig. 4, of which the implementation is based on the non-local model [89] but we extend it to handle the RGB-D fusion. Specifically, the fused feature $Y$ can be written as:

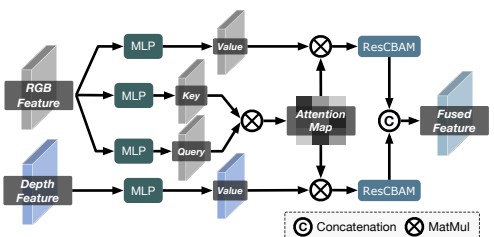

Figure 4: Cross Attention Module (CAM).

$$Y = \mathcal{R}_r(g_r(F_r) \otimes \kappa(F_r)) \parallel \mathcal{R}_d(g_d(F_d) \otimes \kappa(F_r)), \quad (2)$$

where $\mathcal{R}_r$ and $\mathcal{R}_d$ represent two ResCBAMs [90] used for calculating the local attentions for fusion, $\parallel$ is the channel-wise concatenation and $\otimes$ denotes the matmul operation. $F_r$ and $F_d$ indicate the RGB and depth backbone features output by Layer4 (Fig. 2). $\kappa(F_r)$ computes the non-local feature affinity map as $\kappa(F_r) = softmax(\theta(F_r)^T \otimes \phi(F_r))$, where $\theta$ and $\phi$ are learnable linear embedding functions. $g_r, g_d$ are two functions to compute the value features of $F_r$ and $F_d$.

**Geometry Aware Module (GAM).** The GAM aims to enrich the fused features of CAM with high-frequency information for reconstructing geometric details. To this end, we propose to calculate the depth contrasted feature between a local region and its surroundings to capture high-frequency depth variations, as shown in Fig. 5. Specifically, we split out the RGB and depth features: $F_r', F_d'$, according to the channels number, and calculate the contrasted feature for $F_d'$ as:

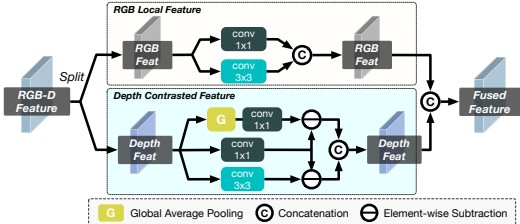

Figure 5: Geometry Aware Module (GAM).

$$C_d = (f_l(F_d') - f_g(F_d')) \parallel (f_l(F_d') - f_l(\mathcal{G}(F_d'))), \quad (3)$$

where $f_l$ denotes the local convolution with a 1x1 kernel and $f_g$ denotes the context convolution with a 3x3 kernel(dilate rate=$x$). $\mathcal{G}$ is a global average pooling operation. $\parallel$ is the concatenation operation. The second item after $\parallel$ represents the local depth feature of the pixels relative to the global feature $F_d'$. As a result, $C_d$ amplifies high-frequency signals at depth transitions, benefiting the prediction of these details (*e.g.*, hairs in Fig. 1). For $F_r'$, we maintain its information through local convolutions.

## 3.2 High-resolution PIFu-Face: $\mathcal{F}_f$

Face regions typically have more high-frequency components than the body (*e.g.*, mouth *v.s.* soft clothing) (Fig. 6(a,b,f)). Even enhanced with depth denoising, the all-in-one PIFu ($\mathcal{F}$) still struggles to represent high-frequency facial details (*e.g.*, nose & mouth in Fig. 6(c,f)). Hence, we propose to express the implicit function $\mathcal{F}$ in a piece-wise manner (*i.e.*, $\mathcal{F}_b$ and $\mathcal{F}_f$) to reduce the complexity of joint occupancy estimation while producing vivid facial and body details. Specifically, we propose to learn the high-frequency facial PIFu representation ($\sigma_f \in [0, 1]$) conditioned on the high-resolution face image ($\mathbf{I}_f$ cropped from the frontal view) and the corresponding denoised facial depth map $\mathbf{D}_{rf}^f$.

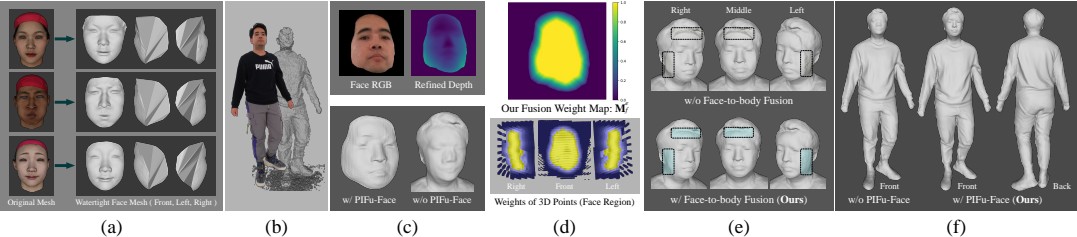

Figure 6: Facial mesh closure on *FaceScape* [94] dataset for training (a). Raw RGBD input images (b). Our reconstructed face model (w/ and w/o PIFu-Face) (c). The weight map $\mathbf{M}_f^e$ and 3D weights for *Face-to-body Fusion* (d). Comparison results on *Face-to-body Fusion* and PIFu-Face (e,f).

**Formulation of $\mathcal{F}_f$.** Given the above inputs (denoted as $\mathbf{T}_f = \{\mathbf{I}_f, \mathbf{D}_{rf}^f, \mathbf{M}_f\}$), we formulate the $\mathcal{F}_f$ along the query point $\mathbf{X}_f \in \mathbb{R}^3$ within the face regions as:

$$\mathcal{F}_f(\mathbf{X}_f, \mathbf{T}_f) = \mathcal{M}_f(\mathcal{B}(\mathcal{H}_f(U^{\uparrow}(\mathbf{T}_f)), \mathbf{x}_f), \mathcal{C}_f(\mathbf{X}_f)) := \sigma_f, \qquad (4)$$

where $\mathcal{H}_f$ denotes the feature extractor for facial images $\mathbf{T}_f$. The function $U^{\uparrow}(\cdot)$ up-samples $\mathbf{T}_f$ to the same resolution as $\mathbf{T}$. $\mathcal{M}_f$ is the implicit function for querying $O_f$. $\mathcal{C}_f(\mathbf{X}_f)$ is defined the same as $\mathcal{C}_i(\mathbf{X})$, but we use the up-sampled masked facial depth $\mathbf{M}_f(U^{\uparrow}(\mathbf{D}_{rf}^f))$ to compute $p_f(\mathbf{X})$.

**Facial Points Selection.** To determine the facial points for inferring $\sigma_f$, we select the points among all the querying points of which the projection $\mathbf{x}_f \in \mathbb{R}^2$ falls inside the bounding-box $\mathbf{R}_f$ of $\mathbf{D}_{rf}^f$ and absolute PSDF value is less than $\alpha$ as $\mathbf{X}_f$ (Fig. 7(b)). We set a flag $v_f(\cdot)$ to mark the facial points as:

$$v_f(\mathbf{X}) = \begin{cases} 1 & \mathbf{x}_f \in \mathbf{R}_f \ \& \ \text{abs}(\mathbf{z}_f - \mathcal{B}(\mathbf{D}_{rf}^f, \mathbf{x}_f)) < \alpha \\ 0 & else \end{cases}. \qquad (5)$$

## 3.3 Face-to-body Occupancy Fields Fusion: $\mathcal{W}$

Simply merging the reconstructed face and body (*i.e.*, replacing $\sigma_b$ with $\sigma_f$ for the facial points) would result in the discontinuity artifacts at the stitching (Fig. 6(e), 1st row). To address this issue, we propose to fuse $O_b$ and $O_f$ via adaptive weights calculated in 3D space. As shown in Fig. 6(d), we compute a 2D fusion weight map ($\mathbf{M}_f^e$) in the $x$-$y$ plane by eroding edges of the facial mask $\mathbf{M}_f$. Along the $z$ axis, we compute the weights through a Gaussian distribution model of the PSDF values. Then, we formulate the joint probability distribution of the final fusion weight $\omega$ as:

$$\omega = \mathcal{B}(\mathbf{M}_f^e, \mathbf{x}_f) \cdot \exp(-\beta \cdot (\mathbb{P}(\mathbf{D}_{rf}^f, \mathbf{X}_f))^2), \quad (6)$$

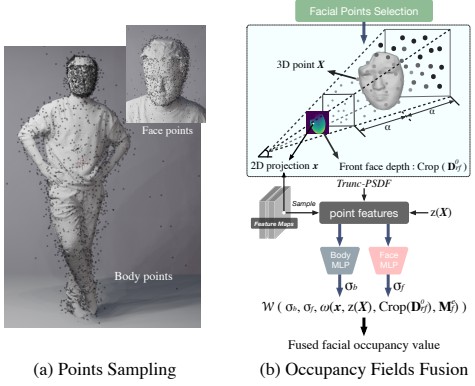

Figure 7: Equal facial and body points sampling (a). Face-to-body fusion (b).

where $\mathbb{P}(\cdot, \cdot)$ denotes the function to calculate the PSDF value, *i.e.*, $\mathbb{P}(\mathbf{D}_{rf}^f, \mathbf{X}_f) = \mathbf{z}_f - \mathcal{B}(\mathbf{D}_{rf}^f, \mathbf{x}_f)$ and the parameter $\beta$ is set to $1e^3$ in default. In Eq. 6, the first term yields smaller values from the center of the face region to its boundaries, which improves the smoothness around the stitching. The second term emphasizes the occupancy values computed by $\mathcal{F}_f$ before and after the face surface. Hence, we can leverage $\omega$ to fuse the $\sigma_b$ and $\sigma_f$ as: $\mathcal{W}(\sigma_b, \sigma_f, \omega) = \omega \cdot \sigma_f + (1 - \omega) \cdot \sigma_b$.

## 3.4 Loss Function

We adopt the extended Binary Cross Entropy (BCE) loss [74] to supervise the predicted occupancy values $\sigma_b$ and $\sigma_f$ on the sampled body and facial points $\tilde{\mathbf{X}} = [\mathbf{X}_b, \mathbf{X}_f]$, which can be written as:

$$L_\sigma = \mu_0 \cdot \sum_{\mathbf{X}_b \in \mathcal{S}_0} \mathcal{L}_B(\sigma_b, \sigma_b^*) + \mu_1 \cdot \sum_{\mathbf{X}_f \in \mathcal{S}_1} \mathcal{L}_B(\sigma_f, \sigma_f^*), \tag{7}$$

where $\sigma_b^*, \sigma_f^*$ are the ground-truth occupancy values for $\mathbf{X}_b$ and $\mathbf{X}_f$. $\mathcal{S}_0$ and $\mathcal{S}_1$ denote the sampled sets. $\mathcal{L}_B$ represents the BCE loss, $\mu_0$ and $\mu_1$ are the weights to balance PIFu-Body and PIFu-Face.

**Regularization Term**. We propose a *Regularization Loss* ($L_{reg}$) to reduce the artifacts in depth-jumping regions during multi-views aggregation, as:

$$L_{reg} = \sum_{\mathbf{X}_b \in \mathcal{S}_j} \mathcal{L}_2(\mathbf{n}(\mathbf{X}_b, \mathbf{T}), \mathbf{n}(\mathbf{X}_b + \epsilon, \mathbf{T})), \tag{8}$$

where $\mathcal{S}_j \in \mathcal{S}_0$ is the set of points projected on the depth-jumping regions (refer to the supplemental for details of obtaining these regions). The parameter $\epsilon$ is a small random uniform 3D perturbation. $\mathbf{n}(\mathbf{X}_b, \mathbf{T}) \in \mathbb{R}^3$ is the normal vector at $\mathbf{X}_b$, defined as $\nabla_{\mathbf{X}_b} \mathcal{F}_b(\mathbf{X}_b, \mathbf{T}) / \|\nabla_{\mathbf{X}_b} \mathcal{F}_b(\mathbf{X}_b, \mathbf{T})\|_2$. Eq. 8 encourages the normals of $\mathbf{X}_b$ to be consistent with those of the points sampled in its neighborhood, hence enhancing the smoothness along the stitching boundaries.

**Depth Denoising Term**. We penalize the per-pixel difference between $\mathbf{D}_{rf}$ and the rendered ground-truth depth map $\mathbf{D}_{gt}$. We also penalize the error of calculated normal maps to prevent $\mathbf{D}_{rf}$ from becoming blurred. The loss for depth denoising is formulated as:

$$L_D = \rho_D \cdot \sum_s^S \lambda_s \mathcal{L}_1(\mathbf{d}_{rf}^s(p), \mathbf{d}_{gt}^s(p)) + \rho_N \cdot \mathcal{L}_2(\mathbf{n}_{rf}(p), \mathbf{n}_{gt}(p)), \tag{9}$$

where $\mathbf{d}_{rf}^s(p)$ denotes the $s$-scale (S=4, the scale of $\psi_g(\mathbf{T})$) predicted depth value of $\mathbf{D}_{rf}$ in pixel $p$. $\mathbf{n}_{rf} \in \mathbb{R}^3$ is the normal vector of $\mathbf{N}_{rf}$ in pixel $p$, where $\mathbf{N}_{rf}$ is the normal map computed from $\mathbf{D}_{rf}$. $\mathbf{d}_{gt}^s(p)$ and $\mathbf{n}_{gt}$ are the corresponding ground-truth value and vector. $\mathcal{L}_1$ and $\mathcal{L}_2$ are the smooth $L1$ loss and $L2$ loss. The weights $\rho_D, \rho_N$ and $\lambda_s$ are used for balancing different loss terms.

The whole loss function can be defined as $L = L_\sigma + \lambda_{reg} \cdot L_{reg} + \lambda_D \cdot L_D$, where the weights $\lambda_{reg}, \lambda_D$ are the balance terms. Refer to the supplemental for detailed implementation information.

## 4 Experiments

**Datasets and Evaluation Metrics**. We use the *THuman2.0* [100] dataset which contains 500 high-quality 3D human scans to train and validate our network. We split the dataset into training and test sets with a ratio of 4:1. We rotate each scan along the yaw axis and render RGBD body portraits at every 6-degree rotation with CUDA acceleration. For PIFu-Face, we pretrain $\mathcal{F}_f$ using the *FaceScape* [94] dataset, which contains 3D head models of different people and expressions. We crop the face regions and make the cropped meshes watertight (Fig. 6(a)). For the input raw depth maps, following [20], we synthesize the sensor noise on $\mathbf{D}_{gt}$ to obtain $\mathbf{D}$. For $\sigma_b^*$ and $\sigma_f^*$, we follow the sampling strategy of PIFuHD [74] to sample body and facial points (Fig. 7(a)), and compute their occupancy values as the ground-truth labels.

We adopt the Point-to-Surface (P2S) distance(cm) and Chamfer distance(cm) for mesh, $L2$ ($1e^{-1}$) and Cosine distance ($1e^{-3}$) for normals as the metrics to measure the errors between the reconstructed and the ground-truth surfaces. Lower metric values indicate better performance.

### 4.1 Comparisons with the State-of-the-arts

We compare our method with six state-of-the-arts human reconstruction methods, including the Multi-view RGBD-PIFu [73], PIFuHD [74], StereoPIFu [40], IPNet [4], DoubleFusion [101] and Function4D [100]. Among them, the DoubleFusion [101] and Function4D [100] are tracking-based, Multi-view RGBD-PIFu [73], PIFuHD (single RGB image) [74], StereoPIFu (stereo RGB images) [40] are PIFu-based, and the IPNet [4] is implicit function based.

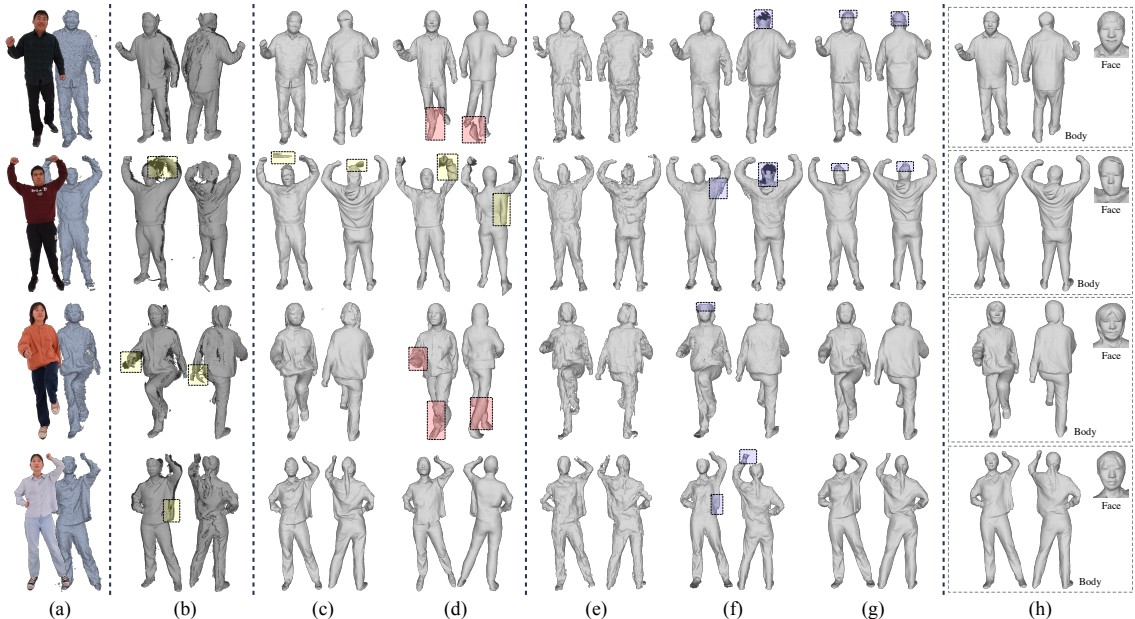

Figure 8: Qualitative comparisons on our captured real data, between the proposed method and five latest state-of-the-art methods. RGBD images of the front view (a). Our refined depths (fused point clouds) (b). Multi-view RGBD-PIFu [73] (c). PIFuHD [74] (d). IPNet [4] (e). DoubleFusion [101] (f). Function4D [100] (g). Ours (h). Zoom in to see the details.

**Quantitative Comparisons.** Tab. 1 reports the quantitative comparisons on our test set between the proposed method and four PIFu (or implicit function)-based approaches. Our method ranks in the first place under all metrics, exceeding the second-best results with a 16.72% reduction in Chamfer and P2S distances, a 9.50% reduction in $L2$ and Cosine distances. The PIFuHD [74] reconstructs 3D human bod-

| Methods | Mesh | | Normal | |
|---|---|---|---|---|
| | P2S$\times 10^{-2}\downarrow$ | Chamfer$\times 10^{-2}\downarrow$ | $L2\times 10^{-1}\downarrow$ | Cosine$\times 10^{-3}\downarrow$ |
| RGBD-PIFu [73] | 0.3335 | 0.3188 | 0.207 | 0.824 |
| PIFuHD [74] | 1.7268 | 1.7423 | 0.512 | 1.576 |
| StereoPIFu [40] | 0.5832 | 0.5425 | 0.328 | 1.193 |
| IPNet [4] | 0.8563 | 0.7247 | 0.196 | 0.751 |
| **Ours** | **0.2652** | **0.2775** | **0.176** | **0.685** |

Table 1: Quantitative comparisons to state-of-the-art methods on our test dataset. The best results are marked in **bold**.

ies from a single RGB image, which cannot handle the topology errors, resulting in the low reconstruction accuracy. The IPNet [4] receives the fused point clouds as inputs, but it still cannot handle the significant noise issues in depths. It also has to fit the SMPL, which may not handle the complex poses. The StereoPIFu [40] can produce reasonable results from the front view but cannot handle the topological errors hidden in other perspectives. Obtaining stereo pairs from multiple views may be a solution, but it significantly increases the computational cost introduced by its 3D voxel features. For RGBD-PIFu [73], we use multi-view RGBDs as inputs to handle the topology errors. However, it still fails to produce reliable results (*e.g.*, artifacts in hairs, over-smoothed faces) when depth maps contain larger noise. In contrast, our method achieves state-of-the-art performance by learning the geometry-aware two-scale PIFu representation.

**Qualitative Comparisons.** Fig. 8 shows the reconstruction results of five existing methods and our method on our captured real data. Although refined depths $\mathbf{D}_{rf}^i$ can be fused (*e.g.*, TSDF-Fusion in [66]) to obtain 3D human models (Fig. 8(b)), the reconstructed results contain large holes and low-quality regions due to sparse inputs and multi-view inconsistencies. The multi-view RGBD-PIFu and the PIFuHD methods often suffer from floating geometry (Fig. 8(c)) and topology errors (Fig. 8(d)) due to non-negligible depth noise and the lack of other view information. For IPNet, even if we take the fused point clouds (from 3 frames) as inputs, the topological errors are still obvious (Fig. 8(e)), and facial details are missing. We also test the DoubleFusion [101] on our captured data. As shown in Fig. 8(f), due to the low quality of the original depths and the sparse embed-graph of the whole body, this volumetric fusion-based approach tends to smooth out regions where we expect to see the high-frequency information (*e.g.*, faces). In addition, when a pose differs significantly from the initial pose, the reconstructed mesh tends to be over-stretched (blue boxes). The Function4D [100] tracks the former and latter frames for the current frame and fuses the multi-frame point clouds to produce

new depth maps. When the motion changes are not small, their method is not easy to reconstruct reasonable high-frequency details (flat faces, hairs in Fig. 8(g)). Our approach leverages the depth denoising to produce robust depth (*e.g.*, hairs in Fig. 8(h)) for the PIFu-based reconstruction process. Moreover, our two-scale PIFu represents the face and body separately to produce vivid details.

## 4.2 Ablation Study

**Multi-task Formulation.** To evaluate the effectiveness of our multi-task design, we compare our geometry-aware PIFu with a sequential model that first performs the depth denoising and then the human reconstruction using two individual networks. The two networks are trained individually. For depth denoising, we use an ablation version of our full model by removing the PIFu-body MLP and the occupancy supervision. For human reconstruction, we remove the depth

| Model | Mesh | | Normal | |
|---|---|---|---|---|
| | P2S$\times 10^{-2}$ ↓ | Chamfer$\times 10^{-2}$ ↓ | $L2 \times 10^{-1}$ ↓ | Cosine$\times 10^{-3}$ ↓ |
| w/o GT Depth | 0.3143 | 0.3085 | 0.213 | 0.801 |
| CAM → AR | 0.2826 | 0.2939 | 0.194 | 0.759 |
| w/o GAM | 0.2695 | 0.2810 | 0.178 | 0.694 |
| w/o GAM & CAM → AR | 0.2901 | 0.2877 | 0.202 | 0.773 |
| w/o $\mathcal{L}_{reg}$ | 0.2760 | **0.2691** | 0.185 | 0.719 |
| **Ours** | **0.2652** | 0.2775 | **0.176** | **0.685** |

Table 2: Ablation study. AR indicates the combination of ASPP [12] and ResCBAM [90]. CAM → AR indicates replacing CAM with AR.

supervision and compute the trunc-TSDF from the input raw depth maps. As shown in Tab. 3, the performance of the sequential pipeline is lower than our model on all five metrics, which illustrates that our geometry-aware features benefit both two tasks. Besides, comparing the refined depths and normals of the sequential model (Fig. 9(b)) with our model (Fig. 9(c)), we find solely depth denoising tends to produce incorrect geometric details (*e.g.*, error depths and normals), as shown in Fig. 9(b,d), which further results in geometric errors (Fig. 9(g)) in the reconstruction process.

| Methods | Mesh | | Normal[from reconstructed mesh] | | Refined Depth |
|---|---|---|---|---|---|
| | P2S$\times 10^{-2}$ ↓ | Chamfer$\times 10^{-2}$ ↓ | $L2 \times 10^{-1}$ ↓ | Cosine$\times 10^{-3}$ ↓ | $L1 \times 10^{-1}$ ↓ |
| Sequential model | 0.3125 | 0.3209 | 0.212 | 0.800 | 0.1261 |
| **Ours** | **0.2652** | **0.2775** | **0.176** | **0.685** | **0.0911** |

Table 3: Quantitative comparisons between the sequential model (depth denoising and 3d reconstruction in sequential) and our proposed geometry-aware PIFu.

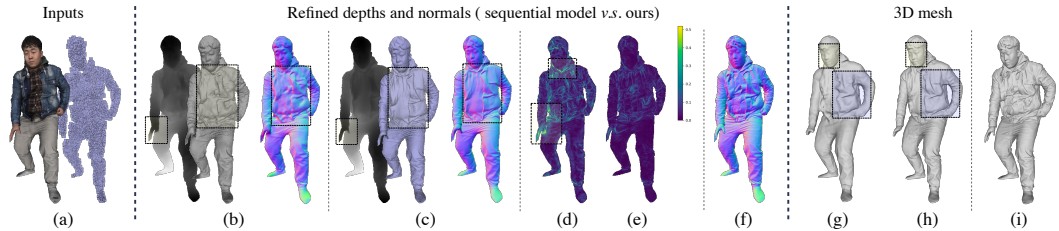

Figure 9: Qualitative comparisons between the sequential model and our proposed geometry-aware PIFu. RGBD inputs of the front view (a). Refined depths and normals of the sequential model (b), and our model (c). Normal error maps between the Ground-truth and the sequential model (d), our model (e). Ground-truth normals (f). Reconstructed 3D results of the sequential model (g) and our model (h). Ground-truth meshes (i). Zoom in to see the details.

**Depth & Occupancy Supervision.** Tab. 2(1st row) shows that by removing the depth denoising task, the performance drops by 14.84% and 18.98% in terms of Mesh and Normal metrics. The decoder $\mathcal{D}(\cdot)$ produces over-smoothed depths and the reconstructed results also lack details, as shown in Fig. 3(e,f) and Fig. 10(c). On the other hand, by removing the 3D supervision, the denoised depth often incorporates incorrect details (*e.g.*, face, hands and clothing regions); See Fig. 3(g) and Fig. 9(b). This experiment further verifies the effectiveness of our multi-task formulation.

**CAM & GAM.** The 2nd-to-4th rows of Tab. 2 evaluate our CAM and GAM modules for RGBD fusion and geometry-aware enrichment, respectively. By replacing the CAM with the common combination of ASPP [12] and ResCBAM [90] (*i.e.*, AR in Fig. 2) or removing the GAM, both the performances drop. We also observe that using AR without GAM yields worse performance on average than removing the GAM. Fig. 10(d,e,f) illustrates the differences, where we can see that these two modules help reconstruct the high-frequency details (*e.g.*, clothes folds of Fig. 10(h,j)).

**PIFu-Face: $\mathcal{F}_f$.** The visual comparisons in Fig. 6(c,f) show that the PIFu-Face, *i.e.*, $\mathcal{F}_f$ significantly improves the ability of our method to reconstruct high-frequency facial details.

**Face-to-body Fusion: $\mathcal{W}$.** The visual comparison in Fig. 6(e) shows that our weighted face-to-body fusion can eliminate the artifacts generated by simply merging the reconstructed face and body.

**Regularization loss:** $L_{reg}$. Fig. 10(g) shows that without the proposed $L_{reg}$, our method tends to produce jagged noise on stitched boundaries where depth may not be consistent.

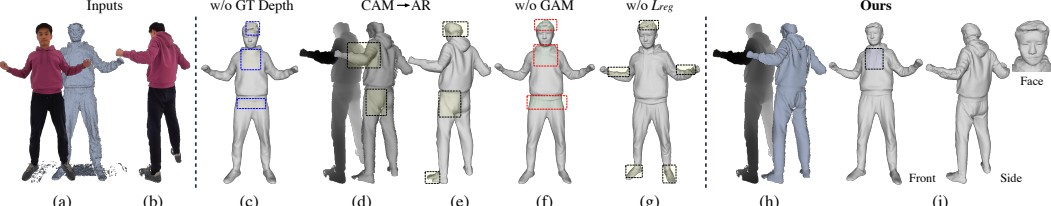

Figure 10: Ablation study. RGBD images of two views (a,b). Reconstruction results and refined depth of ablated versions (c-g). Our results (h,i). Zoom in to see details.

## 4.3  More Experimental Results

**Geometry and Texture**. As shown in Fig. 11, we provide more of our 3d geometric models along with corresponding textured results. We can see that the textured results are of high quality and well aligned with our reconstructed geometries. Moreover, our method is able to handle the loose dressing and respond to the boundary of eyeglasses wearing to some extent (Fig. 11(c,d,e,f)).

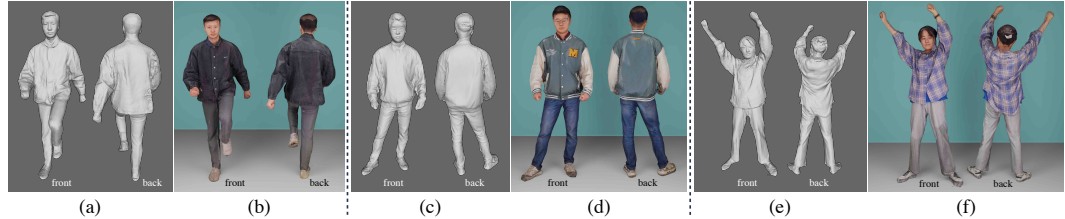

Figure 11: Examples of reconstructed and textured results. 3D geometric models (a,c,e). Textured models (b,d,f). The textured results are generated using MVS-Texturing [86] from the original captured RGBs and rendered via Blender [1].

**Limitation for Hand Details**. As shown in Fig. 12, it remains challenging for our method to generate high-quality hand details when the hand is occluded, or the hand pose is complex. In our two-scale PIFu, we model the face surface from only the front view since the front image contains enough details for the face reconstruction. In contrast, hands have more degrees of movement freedom, which makes them harder to locate and reconstruct independently via extending additional scales based on PIFu from only one view. Combining hand parametric models with our two-scale PIFu may be a feasible solution for hand modeling.

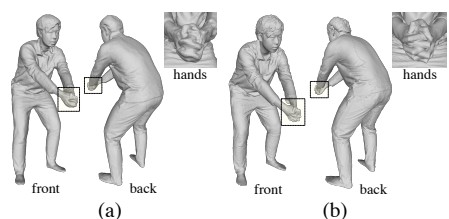

Figure 12: Difficulty for modeling hand details. Ours (a). Ground-truth (b).

## 5  Conclusion

In this paper, we propose the geometry-aware two-scale PIFu representation to reconstruct the digital human body with fine-grained facial details. The first novelty is that we formulate the depth denoising and implicit occupancy estimation in a multi-task learning manner, to exploit their complementary properties. The second novelty lies in that we formulate the two-scale PIFu via two MLPs to represent the face and body separately, to reduce the complexity of modeling high-frequency facial details. Finally, a lightweight face-to-body fusion scheme fuses the face and body occupancy fields to generate reliable reconstruction results of high fidelity. For the limitation of modeling high-quality hand details, we are interested in investigating this topic in the future and exploring a light-weight approach.

**Acknowledgments**: We thank all the reviewers for their constructive comments. Weiwei Xu is supported by NSFC (No. 61732016). This work was partially supported by a GRF grant from RGC of Hong Kong (Ref.: 11205620). This paper is supported by the Information Technology Center and State Key Lab of CAD&CG, Zhejiang University.

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
