# OpenReview forum: "Geometry-aware Two-scale PIFu Representation for Human Reconstruction"
_NeurIPS.cc/2022/Conference — NeurIPS 2022 Accept_

### Official Review · Reviewer_A5r2 · 2022-06-25

**Rating:** 5
**Confidence:** 3
**Soundness:** 2 fair
**Presentation:** 1 poor
**Contribution:** 2 fair

**Summary:**

The paper propose a method for human 3D reconstruction. The main contribution includes: 1. a geometry-aware PIFu method for human reconstruction from RGBD images; 2. a two-scale method dealing with body and face separately and then fusing the results; 3. experiments results showing the effectiveness on Kinect captured data.

**Questions:**

1. Many modules are used in the pipeline. Which of them are completely novel? Which are adapted from existing works?
2. How are the parameters selected?

**Limitations:**

The authors have discussed the limitations. However, it is unclear whether the method works in some challenging scenarios. E.g., loose dressing, eye glasses, etc.

**Strengths And Weaknesses:**

Strengths:

+ The experimental results are good. Clothing deformation and facial features are reconstructed better than compared methods.

+ The ablation study is extensive, showing the necessity of each proposed module.

Weaknesses:

- The method seems to be too complex, and more like stacking existing methods and modules, limiting the novelty of the paper.

- The writing needs significant improvement: (1) The pipeline figure 2 needs to be simplified. Showing too many details in one single figure makes readers get lost. (2) The math should be simplified, too. Instead of defining a lot of symbols, it would be good if the author could describe the key idea (or even for a simplified pipeline) using very simple equations. Complex details could go to the supplementary material. (3) I would not recommend using "\copyright" as the concatenation operation, because it already has other meanings. ‖ could be a better choice.

- Since the method deals with face and body separately, it would be good if the comparison with face reconstruction methods is included in the experiments.

- The related work could be expanded. Here I listed some relevant papers, but the author could include more:
[a] Real-time Deep Dynamic Characters, SIGGRAPH'21;
[b] Texmesh: Reconstructing detailed human texture and geometry from rgb-d video, ECCV'20;
[c] MonoClothCap: Towards Temporally Coherent Clothing Capture from Monocular RGB Video, 3DV'20

---

> ### Author Response · Authors · 2022-08-02
> **Response to Reviewer A5r2**
>
> We thank the reviewer for the constructive comments and address the raised concerns below.
>
> **Q1: The method seems to be too complex, and more like stacking existing methods and modules, limiting the novelty of the paper.**
>
> Our novelty lies in two folds. First, unlike previous methods that typically assume noise-free input, we propose to handle human reconstruction from sparse and noisy RGBD images input by formulating the depth denoising and 3D reconstruction as a multi-tasking learning process. Second, unlike previous methods that typically use all-in-one PIFu to represent the whole body, we propose the two-scale PIFu method to represent the face and body separately, allowing much more details to be reconstructed.
>
> **Q2: The writing needs significant improvement ...**
>
> We thank the reviewer for these helpful suggestions. Will revise our paper accordingly.
>
> **Q3: Since the method deals with face and body separately, it would be good if the comparison with face reconstruction methods is included in the experiments.**
>
> As suggested, we have compared our method to the state-of-the-art face reconstruction method. Visual comparisons can be found in Fig. E of (https://sites.google.com/view/twoscale), which shows that our method produces competitive face reconstruction results. Compared to these methods that explicitly reconstruct the face model, our PIFu-Face implicitly predicts the face occupancy fields, which facilitates our face-body occupancy fields fusion scheme for full-body reconstruction.
>
> **Q4: The related work could be expanded...**
>
> We will cite and discuss the suggested works and include more representative video-based human reconstruction methods.
>
> **Q5: Many modules are used in the pipeline. Which of them are completely novel? Which are adapted from existing works?**
>
> Our method mainly consists of a geometry-aware PIFu-Body branch, a high-resolution PIFu-face branch, and an occupancy fusion scheme.
>
> The geometry-aware PIFu-Body branch is built upon the PIFu [70] representation. We extend it to formulate multi-task learning of depth denoising and 3D reconstruction. Besides, we propose the cross attention module (CAM, see Fig. 4 of our paper) to guide the RGB and depth feature fusion. The CAM module is built upon the non-local model [85] (which was originally proposed for handling RGB inputs), and we extend it to model the non-local correlation across RGB-D inputs. We also propose the geometry aware module (GAM, see Fig. 5 of our paper), which to our knowledge is not built upon any existing works, in order to enrich the high-frequency geometric information of the fused RGBD features.
>
> The high-resolution PIFu-face branch is also built upon the PIFu [70] representation. However, we focus on the estimation of face occupancy conditioned on the high-resolution RGB image and denoised depth information. For our two-scale PIFu, we present a new training method compared to the original PIFu [70] method, as stated in our supplemental (Sec. A).
>
> Our face-to-body occupancy fields fusion, which aims to avoid the stitching artifacts in the implicit occupancy fields, is tailored for our two-scale PIFu method.
>
> **Q6: How are the parameters selected?**
>
> We have discussed implementation details in the supplemental (Sec. A), including the hyper-parameters.
>
> **Q7: The authors have discussed the limitations. However, it is unclear whether the method works in some challenging scenarios. E.g., loose dressing, eye glasses, etc.**
>
> As suggested, we show two examples in Fig. B of (https://sites.google.com/view/twoscale). Results show that our method is able to handle the loose dressing and respond to the boundary of eyeglasses wearing to some extent.

---

> > ### Comment · Reviewer_A5r2 · 2022-08-09
> > **Reply**
> >
> > Thanks for the detailed response. The rebuttal addressed my concerns. I’m leaning towards acceptance.

---

### Official Review · Reviewer_Aogp · 2022-07-11

**Rating:** 5
**Confidence:** 5
**Soundness:** 2 fair
**Presentation:** 3 good
**Contribution:** 2 fair

**Summary:**

This paper presents a 3D reconstruction method of clothed humans from sparse RGBD inputs. This work argues that jointly solving depth refinement and PIFu-based 3D reconstruction is mutually beneficial with empirical evidence. For effective information fusion between RGB and depth, the paper also propose cross attention module and geometry aware module. Additionally, the paper further improves the fidelity of face reconstructoin by processing face and body separately and fused them back into a single 3D model. With THuman2.0 dataset, the proposed approach outperforms existing approaches and the ablation study validates their technical contributions.


**Questions:**

Please provide evidence that multi-task learning is beneficial, not depth refinement itself.

Other comments:
- I would recommend not using different background color for ours in the qualitative results.
- L157: I' and D' do not appear anywhere else.
- Please clarify how to get the input mask.
- L30: minimally clothed
- L297: not small?

**Limitations:**

The paper does not discuss limitations or its societal impact. Discussion would be highly recommended.

**Strengths And Weaknesses:**

The strengths of the paper can be summarized as follows:
- The qualitative results are impressive. Given noisy RGB-D inputs, the proposed method successfully generates high-resolution and complete 3D geometry of clothed humans.
- The paper presents extensive experiments to show the effectiveness of the proposed approach over the existing approaches.
- The proposed rgb and depth fusion module are interesting and shows improvements in performance.

However, there are several weaknesses in this work:
- While the paper argues that the multi task learning of depth refinement and 3D reconstruction is the key contribution, the experiments do not necessarily support this claim. More specifically, it is not clear if solving them together as multi-task is critical or simply providing cleaner depth map is beneficial for the following steps. To validate this, I would recommend running experiments by feeding refined depth that is separately predicted as isolated module into the proposed pipeline. Without such an experiment, the key contribution of this work remains unclear.
- This paper misses one highly relevant paper, JIFF, where PIFu-like model predicts body and face regions separately and fuse them to produce high-quality reconstruction (although they are based on RGB inputs). Given the existence of this work, the contribution of two-scale PIFu would diminish.

JIFF: Jointly-aligned Implicit Face Function for High Quality Single View Clothed Human Reconstruction
Yukang Cao, Guanying Chen, Kai Han, Wenqi Yang, Kwan-Yee K. Wong, CVPR 2022

While the results are impressive, the extent of the paper contributions remain unclear with the current form. Thus, I would stay on the fence, slightly inclined towards rejection.

---

> ### Author Response · Authors · 2022-08-02
> **Response to Reviewer Aogp**
>
> We thank the reviewer for the careful reading and the insightful comments. We are glad to see that the reviewer comments our work has impressive results. We will cite the JIFF paper and revise the paper details. Below we address the raised concerns.
>
> **Q1: While the paper argues that the multi task learning of depth refinement and 3D reconstruction is the key contribution, ... Without such an experiment, the key contribution of this work remains unclear.**
>
> As suggested, we evaluate our multi-task design by using a sequential framework that first performs the depth denoising and then the human reconstruction using two individual networks. The two networks are trained individually. For depth denoising, we use an ablation version of our full model, by removing the PIFu-body MLP and the occupancy supervision. For human reconstruction, we remove the depth supervision and compute the trunc-TSDF from the input depth maps.
>
> We report the errors of mesh (P2S and Chamfer distance), normal (L2 and Cosine distance), and the refined depth map (L1 distance) on our test set in Tab. A of link (https://sites.google.com/view/twoscale). We can see that the performance of the suggested sequential method is lower than that of our method (i.e., multi-task manner).
>
> Comparing the refined depths and normals of the sequential model with our model (Fig. D in https://sites.google.com/view/twoscale), we find solely depth denoising tends to produce incorrect geometric details (e.g., error depths and normals.), which further results in geometric errors in the reconstruction process. This experiment verifies our multi-task design. We will add these quantitative and qualitative results in the final version.
>
> **Q2: This paper misses one highly relevant paper, JIFF [Cao et al. CVPR'22], where PIFu-like model predicts body and face regions separately and fuse them to produce high-quality reconstruction (although they are based on RGB inputs). Given the existence of this work, the contribution of two-scale PIFu would diminish.**
>
> Our two-scale PIFu method differs from the PIFu-like model of JIFF [Cao et al. CVPR'22] in two-folds.
>
> First, JIFF still relies on the 3D morphable face model (3DMM) as face prior. Their face reconstruction capacity is constrained by the 3DMM parameters so that their method may not handle dense facial geometry well (we can see from Fig.4 of JIFF paper that their facial regions are still smoothed). Our method does not rely on 3DMM but leverages high-resolution RGB information to enrich local facial geometry information.
>
> Second, JIFF fuses the face and body MLP features in the feature domain and uses another MLP to produce the whole-body occupancy. This strategy of early implicit fusion may make it challenging for the final MLP to express high-frequency facial details. Our two-scale PIFu uses two MLPs to estimate the face and body occupancy fields, and explicitly fuses the two fields, which assigns each PIFu with more capacity to reconstruct high-frequency details.
>
> We will cite and discuss this paper in our revision. Besides, our work is concurrent to JIFF [Cao et al. CVPR'22]. We did not see this paper when we submitted our paper to NeurIPS 2022.
>
> **Q3: Please clarify how to get the input mask.**
>
> As described in our supplemental (Sec. A), we use the background-matting-v2 [7] method to obtain the body binary masks. For high-resolution facial RGB images, we use the RetinaFace [2] method to detect the front-view face region, and a face skin segmentation network [3] to obtain the facial binary mask.
>
> **Q4: I would recommend not using different background color for ours in the qualitative results. L157: I' and D' do not appear anywhere else. L30: minimally clothed. L297: not small?**
>
> We thank the reviewer for careful reading. Will revise the paper accordingly.
>
> **Q5: The paper does not discuss limitations or its societal impact. Discussion would be highly recommended.**
>
> We have discussed the limitations in Sec.5 of our paper. We observed that our two-scale PIFu method still may not handle hands well. Due to the fact that hands have high degrees of movement freedom in the 3D space, they are difficult to locate and reconstruct from the front view.
>
> We have discussed the societal impacts in our supplemental (Sec. D) that our method may be used to create malicious applications based on 3D reconstruction. We will state them clearly in the revision.

---

> > ### Comment · Reviewer_Aogp · 2022-08-08
> > **Re: Response to Reviewer Aogp**
> >
> > Thanks for the detailed responses and clarification! On the condition that the authors will revise the main paper as promised, I would not be opposed to accepting the paper (raised the score to borderline accept).

---

> > > ### Author Response · Authors · 2022-08-09
> > > **Response to Reviewer Aogp**
> > >
> > > Thank you for the reply! We will carefully revise the main paper according to the comments.

---

### Official Review · Reviewer_7cFn · 2022-07-11

**Rating:** 5
**Confidence:** 5
**Soundness:** 3 good
**Presentation:** 3 good
**Contribution:** 1 poor

**Summary:**

This paper presents a method for reconstructing 3D human models from sparse RGBD inputs. The proposed method is built upon PIFu. To eliminate the negative impact of depth noise, the authors introduce a multi-task learning framework that incorporates the complementary nature of  depth denoising and 3D reconstruction. The authors also introduce an additional MLP for facial regions to further improve the reconstruction quality of human faces. Experiments show that the proposed method can generate high-quality 3D human models from sparse RGBD sensors.


**Questions:**

I am skeptical about the experiments in Fig. 3. In the main text of the paper, the authors emphasis that the task of depth denoising preserves local geometric details while 3D reconstruction provides global topology guidance. However, in Fig. 3 (g), which is the results of solely depth denoising, the refined depth looks much smoother than (h) and does NOT preserve local geometric details. This contradicts with the main insight of this paper and makes me feel that the geometric details is acturally produced in the 3D reconstruction task rather than the depth denoising task. I think the authors should discuss more about the experiment in Fig. 3 and provide more results if possible.



**Ethics Review Area:**

["I don’t know"]

**Limitations:**

The authors have discussed the limitations in Sec. 5 and the potential societal impact in the supplement material. No more improvement is needed.



**Strengths And Weaknesses:**

Strengths:

* The results look impressive. Given noisy depth measurement from sparse RGBD sensors, the proposed method is able to reconstruct clean 3D models with high-quality geometric details.

* The main contributions of this paper are convincingly validated. The authors present extensive experiments throughout the paper.

* This paper is overall well-written and easy to follow.


Weaknesses:

* The novelty of this paper is a bit of weak. In the introduction, the authors claim the two-scale PIFu representation as one technical contribution. However, a similar design has already been proposed in PIFuHD. I understand that the fine-scale network in PIFuHD is still designed for the whole body while the fine-scale MLP in this paper only focuses on the facial region, but I would regard this difference as engineer choices instead of a "technical contribution".

* No textured results are demonstrated. The authors are encouraged to present the textured models because it is a more intuitive way to evaluate the geometric reconstruction accuracy.

* To evaluate the multi-task design, the authors simply remove the depth denoising task. I think a better baseline for this experiment would be a sequential framework where the depth maps are firstly refined by the depth denoising module and then fed into the 3D reconstruction network.

* Missing citation: Zheng et al. PaMIR: Parametric Model-Conditioned Implicit Representation for Image-based Human Reconstruction. IEEE T-PAMI.

******************************************************************
Final comments:
Thanks for the authors' response which addresses some of my concerns. After reading other reviews and the rebuttal, I would like to keep my initial rating. The authors should add their additional experimental results into the final paper if the paper gets accepted finally.

---

> ### Author Response · Authors · 2022-08-02
> **Response to Reviewer 7cFn**
>
> We are glad to see the reviewer comment on our work achieving impressive results. We will cite the PaMIR paper. In the following, we address the raised concerns in the review comments.
>
> **Q1: The novelty of this paper is a bit of weak. ... I would regard this difference as engineer choices instead of a "technical contribution".**
>
> The PIFuHD [71] method extends PIFu [70] representation in a coarse-to-fine manner. Hence, PIFuHD [71] still suffers from lacking correct facial details. Our motivation for designing two-scale PIFu is to reduce the complexity of full-body occupancy fields estimation by modeling face and body separately. This strategy further allows our method to leverage the existing high-resolution face dataset (e.g., FaceScape [89] ) to model high-frequency details in the face part.
>
> **Q2: No textured results are demonstrated. The authors are encouraged to present the textured models because it is a more intuitive way to evaluate the geometric reconstruction accuracy.**
>
> We thank the reviewer for this suggestion. Textured results are provided in Fig. A and Fig. B via link (https://sites.google.com/view/twoscale). Here we used the MVS-Texturing [Waechter et al. ECCV'14] to generate textured meshes from the original captured RGB images. It can be seen that our generated texture results are of high quality and well aligned with our reconstructed geometries. We will add them in the revision.
>
> **Q3: To evaluate the multi-task design, the authors simply remove the depth denoising task. I think a better baseline for this experiment would be a sequential framework where the depth maps are firstly refined by the depth denoising module and then fed into the 3D reconstruction network.**
>
> As suggested, we evaluate our multi-task design by using a sequential framework that first performs the depth denoising and then the human reconstruction using two individual networks. The two networks are trained individually. For depth denoising, we use an ablation version of our full model, by removing the PIFu-body MLP and the occupancy supervision. For human reconstruction, we remove the depth supervision and compute the trunc-TSDF from the input depth maps.
>
> We report the errors of mesh (P2S and Chamfer distance), normal (L2 and Cosine distance), and the refined depth map (L1 distance) on our test set in Tab. A of link (https://sites.google.com/view/twoscale).  We can see that the performance of the suggested sequential method is lower than that of our method (i.e., multi-task manner).
>
> Comparing the refined depths and normals of the sequential model with our model (Fig. D in https://sites.google.com/view/twoscale), we find solely depth denoising tends to produce incorrect geometric details (e.g., error depths and normals.), which further results in geometric errors in the reconstruction process. This experiment verifies our multi-task design. We will add these quantitative and qualitative results in the final version.
>
> **Q4: I am skeptical about the experiments in Fig. 3. ... I think the authors should discuss more about the experiment in Fig. 3 and provide more results if possible.**
>
> Since the input RGB images for the depth denoising task contain edges and object boundaries which are clues to recovering the local geometric details, we observed that the denoised results had such local details. However, there might exist errors in the details reconstructed by the denoising task. For example in Fig. 3 (g), although the facial region looks sharper than that in Fig. 3 (h), some local geometric details are incorrect. These errors can be further mitigated by the 3D reconstruction task.  Since our Geometry-aware Features are used for both depth denoising and 3D reconstruction tasks, the two tasks can benefit each other to further improve the details of reconstruction and refined depth, as shown in Fig. 3(h, j, f). Moreover, the 3D reconstruction task also provides global topology guidance for the occupancy fields. To avoid possible confusion, We plan to add a description after Line 50 in our paper to explain that the 3D reconstruction task can work with denoising to further improve the quality of the reconstruction results.
>
> Besides, we show more qualitative results in the link (https://sites.google.com/view/twoscale). As shown in Fig. D (b,c,d), solely depth denoising tends to introduce incorrect details, resulting in larger errors for normals compared to our multi-task model. We will add these qualitative results in the revision.
>
> ---
>
> MVS-Texturing : Waechter M, Moehrle N, Goesele M. Let there be color! Large-scale texturing of 3D reconstructions[C]//European conference on computer vision. Springer, Cham, 2014: 836-850.

---

### Official Review · Reviewer_wHR5 · 2022-07-12

**Rating:** 6
**Confidence:** 3
**Soundness:** 4 excellent
**Presentation:** 4 excellent
**Contribution:** 3 good

**Summary:**

This paper proposed several additions onto the PIFU architecture that makes it perform better when trained and evaluated with noisy sparse RGBD views. The paper has three major contributions. 1.  A two scale PIFU model to render detail at different scales and to focus on the face of the human avatar. 2. An extension of PIFU to take in noisy RGBD data from real world sensors. 3. A demonstration of the results on 3 Kinects

**Questions:**

* Some of the text on the figure is so small it can be hard to read when written out.
* In Figure 9, what does "Ours" have a different background
* For the limitations section, did the authors try to train a 4 scale occupancy model with seperate occupancy models to model each hand? It sounds like the authors may have experimented with doing so, but failed to get good results. If so, this information would be very helpful to at least include in the paper's supplement.

**Limitations:**

Yes

**Strengths And Weaknesses:**

The paper is very well written. It provides competitive qualitative and quantitative results with state of the art. The paper is very well written and include useful figures and diagrams. One useful novel contribution their method includes merging the occupancy maps of the face and body. I think this might be the most general contribution in the paper as it stands. The separate modeling of the face and body with their own occupancy fields might also be a good way to optimize the number of parameters needed for high fidelity body construction. The approach is just taking some rather straightforward impovements onto the existing PiFU HD, so the technique itself isn't particularly original. However, it does define an interesting use case for PiFU, sparse RGBD reconstruction with Kinects and demonstrates noticeable improvements on it. I will note that I am not extremely familiar with the latest literature surrounding PiFU so may be missing some related work here, but the the contributions are somewhat novel and useful to future work on digital humans.

One weakness is the lack of qualitative examples describing the weaknesses noted in Section 5. It would be nice as a reader for the paper to demonstrate visually why hands remains challenging.

---

> ### Author Response · Authors · 2022-08-02
> **Response to Reviewer wHR5**
>
> We are delighted to see that Reviewer wHR5 comments on our work defining an interesting case for PIFu. We will revise our figures and font sizes accordingly and address the raised concerns below.
>
> **Q1: The approach is just taking some rather straightforward improvements onto the existing PiFU HD, so the technique itself isn't particularly original.**
>
> Our method and PIFuHD [71] both use two PIFu-based modules to predict the occupancy fields. However, our method is fundamentally different from PIFuHD in two folds.
>
> First, PIFuHD [71] designs two-scale PIFus in a coarse-to-fine manner, where the face and body occupancy fields are still jointly estimated. Hence, PIFuHD suffers a similar problem of the incorrect flat facial surface as that of PIFu [70]. Our two scale-PIFu uses two MLPs to estimate the face and body occupancy fields separately, which assigns each PIFu with more capacity to reconstruct high-frequency details.
>
> Second, unlike the PIFuHD [71], we aim to address the problem of human reconstruction from sparse noisy RGBD sensors. Accordingly, we formulate depth denoising and 3D reconstruction as a multi-task learning process.
>
> **Q2: One weakness is the lack of qualitative examples describing the weaknesses noted in Section 5. It would be nice as a reader for the paper to demonstrate visually why hands remain challenging.**
>
> We thank the reviewer for this suggestion. Two visual examples are provided in Fig. C via link (https://sites.google.com/view/twoscale). We can see that the geometry-aware two-scale PIFu still struggles to reconstruct high-quality hand details when the hand is occluded or the hand pose is complex. In addition, the large depth noise and less RGB information for hands will also make it challenging to obtain high-frequency hand details.
>
> **Q3: For the limitations section, did the authors try to train a 4-scale occupancy model with separate occupancy models to model each hand?**
>
> We haven't successfully trained a 4-scale PIFu model to reconstruct each hand additionally. In our two-scale PIFu, we model the face surface from only the front view RGBD since the front image contains enough details for the face reconstruction.
>
> In contrast, hands have more degrees of movement freedom in the 3D space, which makes them harder to locate and reconstruct from only one view based on PIFu [70]. Furthermore, a high-quality hand dataset needs to be prepared to pre-train the 4-scale PIFu, which is also a time-consuming task.
>
> In another way, combining hand parametric models with our two-scale PIFu may be a feasible solution for hand modeling, and we will explore the topic in the future.

---

> > ### Comment · Reviewer_wHR5 · 2022-08-08
> > **Reply**
> >
> > Thank you for the detailed rebuttal. There is some interesting discussion here and visuals here about why modeling the hands separately does not improve the quality as of now. This poses the questions about what other body parts may benefit from this separate modeling processing. For instance, would modeling the hair/scalp separately also prove beneficial? Are the improvements due to better utilization of the parameters in areas that are more geometrically complex (like the face / head)? If so, could this approach be further generalized to dynamic capture of non-humanoid shapes? Answering these questions would make for interesting follow up work.

---

> > > ### Author Response · Authors · 2022-08-09
> > > **Reply to Reviewer wHR5**
> > >
> > > Thanks for the interesting questions. Our two-scale PIFu representation has the following two working prerequisites. First, the independently modeled regions (e.g., the face regions) are salient and easy to segment. Second, for these regions, there are high-quality datasets (e.g., FaceScape [89] dataset we used for modeling face region) available for the independent pre-training.
> > >
> > > **Q1: For instance, would modeling the hair/scalp separately also prove beneficial?**
> > >
> > > Yes, modeling the hair/scalp separately is beneficial. We observe that the depth noise of the Kinect camera is large (sometimes the depth value is missing) in the hair/scalp regions. Hair also has fine structures that are error-prone to noise. Hence, our method utilizes depth denoising to handle the noise issue. On the other hand, the hair regions are relatively easy to locate/segment compared to the hand regions; our two-scale PIFu can be further extended to model the high-frequency details of hair/scalp independently (using a high-quality hair dataset for pre-training).
> > >
> > > In addition, some issues remained to consider when we extend our method to model hair (as well as other body parts) separately. First, by modeling more regions/parts separately, one accompanying challenge is the increased computational cost. Second, with more regions being modeled independently, we may need to determine a better occupancy fields fusion strategy to avoid stitching artifacts.
> > >
> > > **Q2: Are the improvements due to better utilization of the parameters in areas that are more geometrically complex (like the face / head)?**
> > >
> > > Yes, better utilization of parameters in geometrically complex areas is important to the final reconstruction performance. Face/head regions not only contain geometrically complex details but are key factors when we assess the fidelity of human reconstruction. By assigning an individual PIFu for face/head, our method can leverage high-resolution facial input images and the pre-training on the FaceScape dataset for high-fidelity reconstruction.
> > >
> > > **Q3: Could this approach be further generalized to dynamic capture of non-humanoid shapes?**
> > >
> > > We thank the reviewer for this interesting question. First, although in this paper our two-scale PIFu method reconstructs the human body frame by frame, it is possible to incorporate temporal constraints to handle the dynamic capture of targets. Second, our two-scale PIFu representation is not limited to modeling human bodies. Hence, it is possible for our method to handle the dynamic capture of non-humanoid shapes.
> > >
> > > However, there will be a few questions to answer when we apply our method to non-humanoid shapes. For example, we will need to determine whether it is necessary to use the two-/multi-scale PIFu representation (considering the computational cost), and which regions are necessary to model separately. For animals, we probably will need to model their faces/heads separately. For non-humanoid robots, we may probably focus on their functioning parts.

---

### Author Response · Authors · 2022-08-08
**Response to Reviewers**

Dear all reviewers,

We sincerely thank you for the previous review time and constructive comments. We have provided additional results and explanations of the issues, including our novelty compared to PIFuHD and JIFF, validation of the multi-task designs, textured results, comparison with face reconstruction methods, etc, which we believe have addressed your main concerns. We will revise our paper accordingly and further improve our supplementary materials. If you have any further questions, please let us know.

Thanks,

Authors

---

### Comment · Area_Chair_novw · 2022-08-08
**Any thoughts from reviewers?**

Hi Reviewers,

The discussion period is closing soon. Please take a look at the responses from the authors. If you have further questions, please ask them now, since the authors will be unable to respond soon. It's substantially more productive, effective, and reasonable to have a quick back-and-forth with authors now than to raise additional questions or concerns post-discussion period that the authors are unable to address.

Thanks,

AC

---

### Meta-Review · Area_Chair_novw · 2022-08-22

**Recommendation:** Accept
**Confidence:** Certain

**Metareview:**

All reviewers were in favor of acceptance. The AC examined the paper, reviews, and author response, and is inclined to agree. The AC would encourage the authors to incorporate their responses to the reviewers into the final version of the paper.

**Award:**

No

---

### Decision · Program_Chairs · 2022-09-14

Accept